# Application and Validation of a Dynamic Energy Simulation Tool: A Case Study with Water Flow Glazing Envelope

**Belen Moreno Santamaria** [1], **Fernando del Ama Gonzalo** [2,*], **Danielle Pinette** [2], **Roberto-Alonso Gonzalez-Lezcano** [3] , **Benito Lauret Aguirregabiria** [1] and **Juan A. Hernandez Ramos** [4]

[1]  Department of Construction and Architectural Technology, Technical School of Architecture of Madrid, Technical University of Madrid (UPM), Av. Juan de Herrera, 4, 28040 Madrid, Spain; belen.moreno@upm.es (B.M.S.); benito.lauret@upm.es (B.L.A.)

[2]  Keene State College, 229 Main St, Keene, NH 03435, USA; pinetteda@gmail.com

[3]  Escuela Politécnica Superior, Universidad San Pablo-CEU, Montepríncipe Campus, Boadilla del Monte, 28668 Madrid, Spain; rgonzalezcano@ceu.es

[4]  Department of Applied Mathematics, School of Aeronautical and Space Engineering, Technical University of Madrid (UPM), Plaza Cardenal Cisneros 3, 28040 Madrid, Spain; juanantonio.hernandez@upm.es

*  Correspondence: fernando.delama@keene.edu

**Abstract:** The transparent materials used in building envelopes significantly contribute to heating and cooling loads of a building. The use of transparent materials requires to solve issues regarding heat gain, heat loss, and daylight. Water flow glazing (WFG), a disruptive technology, includes glazing as part of the Heating, Ventilation and Air Conditioning (HVAC) system. Water is transparent to visible wavelengths, but it captures most of the infrared solar radiation. As an alternative to fossil fuel-based HVAC systems, the absorbed energy can be transferred to the ground through borehole heat exchangers and dissipated as a means of free-cooling. Researchers of the Polytechnic University of Madrid have developed a software tool to calculate the energy balance while incorporating the dynamic properties of WFG. This article has studied the mathematical model of that tool and validated its ability to predict energy savings in buildings, taking spectral and thermal parameters of glazing catalogs, commercial software, and inputs from the measurements of the prototypes. The results found in this article showed that it is possible to predict the thermal behavior of WFG and the energy savings by comparing the thermal parameters of two prototypes. The energy absorbed by the water depends on the mass flow rate and the inlet and outlet temperatures.

**Keywords:** water flow glazing; building energy simulation; experimental validation

## 1. Introduction

The percentage of glazing in buildings affects significantly heat gains and losses [1,2]. The visual and thermal properties of glass, the fluid used to fill the space between glass panes, and the use of coatings alter the energy absorption in each layer of the glazing [3]. Article 6 of the Energy Performance of Buildings Directive (EPDB 2010) states that building designers have to consider renewable sources of energy for Heating, Ventilation and Air Conditioning (HVAC) and high-performance envelopes [4]. The first step to accomplish the goal of zero energy buildings is to minimize the heating loads. The second step is to select and size innovative and more efficient devices for heating and cooling, and the third step includes the use of renewable energies [5].

### 1.1. Closed-Loop Ground Source Systems

Borehole heat exchangers have been introduced as an alternative to fossil fuel-based HVAC systems in Europe and North America over the last decades [6]. Open-loop systems use groundwater as a geothermal heat source and may affect local temperature evolution [7]. Closed-loop systems use a fluid that circulates through borehole heat exchangers to transfer the thermal energy from the subsurface to a hydronic HVAC system. Some articles report tests on closed-loop borehole heat exchangers and their capability to extract energy from the ground in winter. A refrigerant fluid transports the heat. This solution is optimum in cold countries when the systems allow heat injection into the ground in summer [8,9]. At a certain depth, the temperature of the ground remains steady over the year, so borehole heat exchangers are considered an optimum solution when the available area is limited [10]. Plastic pipes made of polyethylene or polypropylene are deployed in the boreholes and the irregular space that remains between the pipe and the surrounding soil has to be filled with bentonite, a fluid that absorbs water and expands. This property enhances heat exchange and performance as a thermal conductor [11,12]. Ground Source Heat Pumps (GSHP) exploit the nearly constant ground temperature over the year to increase their performance coefficients. GSHP fit well in the low-CO2 energy supply system [13]. The soil mass is considered a great means of thermal energy storage, and the temperature oscillations are reduced at 10-20 m depth over the seasons. [14].

### 1.2. The Water Flow Glazing (WFG)

The Architectural Engineering and Construction (AEC) industry has focused on the building envelope performance to reduce the heating and cooling loads. The elements used in the building envelope contribute decisively to the energy performance of buildings. Using transparent materials requires first, to understand their spectral properties and second, to develop systems that can solve some of the issues regarding heat gain, heat loss, and daylight [15]. Solar heat gain from infrared radiation is the factor that contributes the most to the cooling loads of a building [16]. Water flow glazing (WFG) is a disruptive technology that includes glazing as part of the HVAC system [17]. WFG includes a circuit that allows fluid to flow through space between two glass panes. In a double glazing, the transmissivity coefficient depends on the radiation's wavelength. Clear glass is transparent to visible and near-infrared (NIR) wavelengths. However, water is opaque to NIR wavelengths, while its visible transmittance is very high [18]. Therefore, water captures solar NIR and increases its temperature through the window. WFG reduces solar gains through transparent envelopes without hindering light transmission. The flow of the water enables the building envelope to apply energy-saving measures, such as energy storage in buffer tanks, and facade homogenization [19].

The lack of thermal inertia affects the thermal behavior of glazing [20] because the daily oscillations in the outdoor temperature are replicated indoors. The thermal inertia buffers the daily fluctuations in the outside temperature. By decreasing these oscillations, the HVAC systems need less power. In warm dominant climates, variable thermal inertia enables better management of the energy flow to and from the building [21,22]. Using WFG, the strong sunlight of the summer is kept out of the building. Releasing the excess of heat in the ground during the summer allows to increase the thermal inertia of a light and transparent envelope. Therefore, the regulation of the glazing's thermal inertia can be achieved by changing the flow rate that circulates through the panels. The lower the flow rate, the lesser the damping effect.

Active water flow glazing raises its thermal inertia compared to conventional double or triple glazing. The water inside is in continuous movement as it manages to release the absorbed heat. Therefore, it reaches to thermal inertia comparable to other traditional construction elements (brick and concrete walls) by simply connecting the system to energy sinks, such as borehole heat exchangers or buffer tanks.

### 1.3. Dynamic Building Energy Modeling

Accurate prediction of the behavior of glazing facades has to include the quality of construction and permeability to the passage of air and vapor. Those aspects depend on the design skills, as much

as on the quality of the construction. The dynamic thermal performance of an envelope produces variations in thermal properties over the day, such as variable thermal inertia, which the Fourier model does not take into account [23]. Thus, developing and validating new mathematical models that can simulate dynamic properties is an important goal. Energy Plus is the most popular engine for calculating energy load and energy system performance of buildings [24,25], but it does not include a module to simulate WFG.

### 1.4. Innovation and Objectives

Engineers and researchers of the Polytechnic University of Madrid have developed a tool to calculate preliminary energy balances of a building. Under the two main tabs, thermal behavior and sun energy, the interface allows the introduction of parameters, such as glass types, coatings, fluids filling the space between glass panes, thermal mass of the building, and internal heat gains. The software library is an open-source project and can be used by developers to validate results. According to some studies, architects and building designers are prone to use simple energy simulation tools than others that might include more options but are more complicated [26,27].

The transient state of the building envelopes is affected by changes in temperature and solar radiation. The steady-state model is not accurate when it comes to dynamic forms of heat transfer, additionally affected by indoor HVAC systems. Remote sensing systems can be used in future studies to compare the actual energy performance with the results of simulation models [28,29]. Other technologies, such as switchable glazing, can adapt to different weather conditions. Polymer dispersed liquid crystal (PDLC) devices can adapt to different weather conditions, but their expected range of U value and g-factor is not as broad as expected for WFG [30]. Suspended Particle Devices (SPD) switches from transparent to colored due to the suspension of freely arranged particles between two glass panes. The electrochromic (EC) glass varies transmission and reflection parameters by electrical stimuli to mobile ions in the electrochromic layer. The solar heat gain coefficient of SPD and EC ranges from 0.5 in the transparent state to 0.06 in the non-transparent state [31]. The addition of double glazing to an SPD can vary the U value from 2.98 W/m$^2$ K to 1.99 W/m$^2$ K [32]. The heat transfer coefficient of SPD ranges between 5.02 W/m$^2$ K and 5.2 W/m$^2$ K for opaque and transparent states [33]. The critical aspects remain in the high cost of the products and a lack of standardization in the manufacturing process [34].

This paper aims to study the dynamic thermal parameters of WFG. The prototype with WFG in facades and the roof, connected to borehole heat exchangers, was built. The only additional energy source was the electricity needed to run the circulation water pumps. The flow of water requires a circulating water pump that consumes electric energy, though the consumption of that pump is low compared with the heating and cooling savings over the year. Some articles show that photovoltaic panels integrated into the WFG facade can provide the water pump with energy [35]. A second prototype was made with the same dimensions and double glazing with an air cavity.

There were three main objectives in the analysis of the WFG prototype. First, it allowed comparing the indoor temperatures of the WFG cabin and the Reference cabin. Second, the simulation tool based on the mathematical model to predict the performance of WFG was validated using real data. Finally, the effect of the borehole heat exchangers connected to the WFG cabin and its influence on the heat gains in the water chamber were taken into account. By comparing the indoor temperatures of two prototypes, the influence of WFG as a means of energy manager was tested. Not only has the indoor temperature in both prototypes been analyzed, but also the capability of WFG to absorb energy with a specific mass flow rate. The results of this study show that the solution of double glazing with a circulating water chamber is a less polluting and more efficient option than the systems currently used, which are, most of them, fed by fossil fuels. This study aims to validate a new building energy simulation tool by comparing numerical results and real data obtained in test rooms. The empirical tests were carried out over a year in a free-floating indoor temperature regime with variable outdoor conditions.

The challenges of WFG are related to the ability to produce large panels and their behavior in the long term. Although considerable efforts have been made in the industrialization process, there is still low market penetration because of the limited information between professionals and consumers. It is not the aim of this article to study the efficiency of closed-loop shallow geothermal systems. Data from the borehole heat exchangers have been used as input to validate the results of the mathematical model.

## 2. Materials and Methods

Since commercial building energy simulation tools do not include WFG as an option for energy management, it is necessary to validate the outputs of simulation with data from real prototypes. The first subsection described the geometry, the energy management, and the materials used in two cabins. The second one showed the equations and the mathematical models of the simulation tool. Finally, the last subsection set the criteria to select the spectral and thermal parameters of the WFG.

### 2.1. Description of the Prototype

For this prototype, active water flow glazing is combined with borehole heat exchangers. On one hand, the active envelope circulates the water at a temperature close to comfort conditions with little energy cost, making the most of the ground's thermal inertia. Another beneficial effect is that the ground temperature increases during the summer as heat absorbed by the water flowing, through the facades and roof, is transferred to the ground by the borehole heat exchangers.

The prototype is located near Madrid, Spain (latitude 40°36′42″ N, longitude 2°26′57″ O, altitude 1111 MAMSL). The dimensions of the prototypes are 2 m × 2 m × 2 m, with a total volume and floor area of 8 m$^3$ and 2 m$^2$, respectively. The prototypes are made of aluminum frames without thermal breakage. The North facades consist of sandwich panels with a thermal transmittance of U = 0.6 W/m$^2$ C. The makeup of the panels is: 1 mm of aluminum, 5 cm extruded polystyrene, 1 mm aluminum. The floor of the prototype is constructed with the same panels as the North facade. The first prototype cabin incorporates WFG. This prototype will be referred to as WFG cabin. While the second prototype cabin, of the same dimensions, orientation, and glazing, has an air chamber in lieu of WFG. This prototype will be referred to as Reference cabin (RC). For the WFG prototype, four borehole heat exchangers are buried 50 m below the surface. There are two different closed-circuit circulating systems. For the primary circulating system, pipes with refrigerant fluid connect from the borehole heat exchangers to the circulating pump. There are two secondary systems, one for the WFG on the vertical facade and another for the WFG on the horizontal roof. For this circulating system, water connects the circulating pump to the WFG. Figure 1 represents the schematics of the energy management system of the WFG cabin. The primary circulating system goes from the plate heat exchangers (1) to the borehole heat exchangers (4). The two secondary circulating systems go from the circulating pump (2) to the WFG facade (3). Both secondary circuits contain a precision flow meter and a meter for monitoring the inlet and outlet temperature.

Calculating the required length of the borehole heat exchangers depends upon the geology and the thermal conductivity of the soil [16]. Typical values for specific heat extraction range between 40 and 70 watts per meter of borehole length.

Weather data over the year were recorded and compared with the building energy simulation (BES) model. The indoor temperature sensors were placed below the roof frame in both prototypes. The outdoor and indoor air temperatures were monitored by several thermocouples (error ±0.5 °C). A pyranometer with a spectral response from 280 to 2800 nm, a linear response up to 3000 W/m$^2$, and an absolute error of 3% measured the solar radiation on the horizontal roof.

Figure 2 shows the real prototypes on location. The climate of the location of the prototypes experiences severe seasonal changes in temperature. The cabin on the left is the WFG cabin and the right is the Reference cabin (RC).

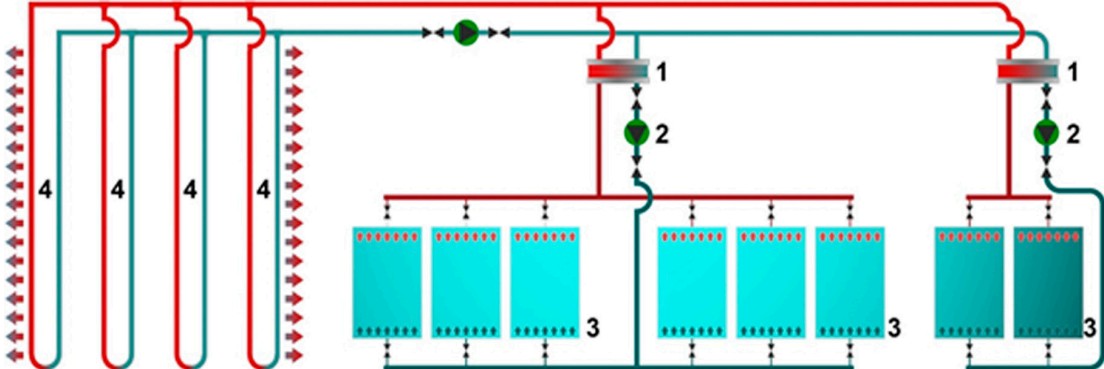

**Figure 1.** Schematics of the energy system. 1. Plate heat exchangers, 2. Water circulation pumps and precision flow meters, 3. Water flow glazing (WFG), 4. Borehole heat exchangers.

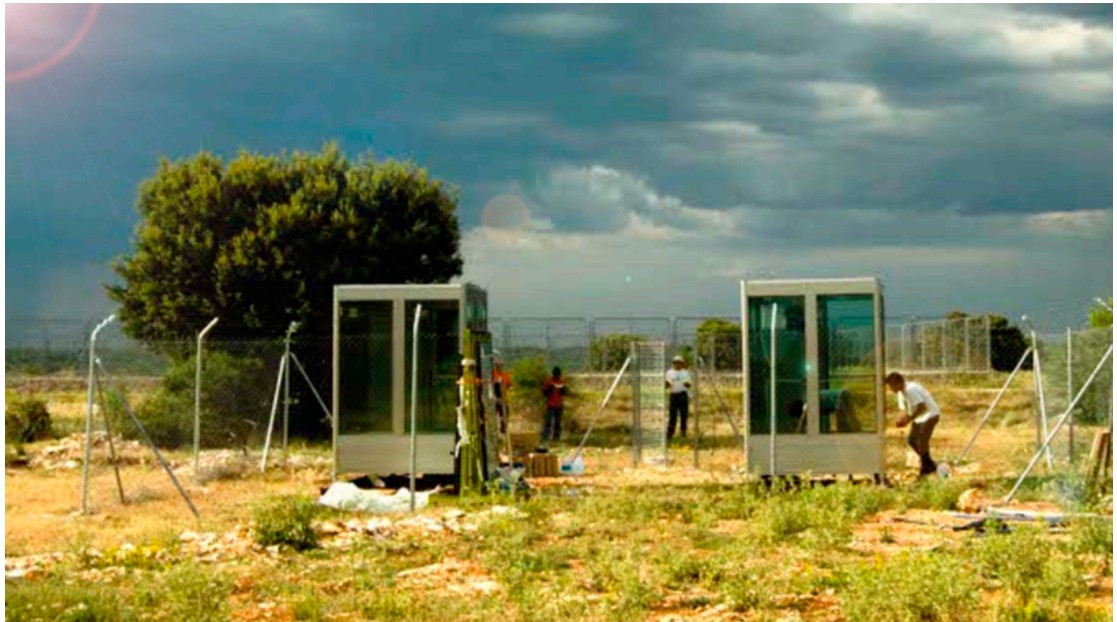

**Figure 2.** Real prototypes on their location.

### 2.2. Mathematical Model

Previous studies have shown a mathematical model that explains the behavior of WFG [36–38]. This study used the equations and the mathematical models from those articles, along with data from commercial software, to assess the performance of the prototypes described in Section 2.1. The internal temperature of the prototype obtained experimentally was compared with the results from the mathematical model. Commercial catalogs provided thermal and spectral properties of glazing.

#### 2.2.1. Double Glazing with a Gas Cavity

Heat flux inside double glazing with a gas cavity is caused by the direct and diffuse solar radiation, $i_0$, and the difference between the indoor and outdoor temperatures, $(\theta_e - \theta_i)$. The thermal transmittance, $U$, is the parameter that explains the heat transfer. The units of measurement are W/m$^2$ K, and it is determined by the European Standard [39,40]. Two components explain the value of the g factor. The first one is the direct solar transmittance, T. The second one is the fraction of the solar radiation absorbed by the glazing, which is later transferred inside by natural convection. Equation (1) represents the heat flux that goes through the glazing. The heat flux $q$ depends on $U$ and $g$ [41].

$$q = U(\theta_e - \theta_i) + gi_0, \tag{1}$$

the thermal transmittance, $U$, and the $g$ factor are defined by:

$$\frac{1}{U} = \frac{1}{h_e} + \frac{1}{h_i} + \frac{1}{h_g},$$ (2)

$$g = T + A_I,$$ (3)

$$A_I = U\left[A_1\left(\frac{1}{h_e}\right) + A_2\left(\frac{1}{h_e} + \frac{1}{h_g}\right)\right],$$ (4)

where $h_i$ is the interior heat transfer coefficient, $h_e$ is the exterior heat transfer coefficient, $h_g$ is the heat transfer coefficient of the air chamber, and $A_1$ and $A_2$ are the thermal absorptances of the exterior and the interior glass panes, respectively.

### 2.2.2. The Influence of Water Flow Glazing

With a fluid circulating between two glass panes, there are more parameters to take into account to evaluate the thermal performance. The temperature and the mass flow rate of the fluid are variable and this affects the heat flow through the glazing. The result of adding the transmitted solar energy radiation $(T\,i_0)$ and the secondary internal heat flux $q_I$ is the total indoor heat flux, $q$

$$q = T\,i_0 + q_I.$$ (5)

A general solution of heat flux for WFG is given in Reference [36], adequate for all glazing configuration, is presented below. The algebraic equation is valid, assuming that the mass flow rate is uniform, the values for convective and radiative coefficients do not change with time, and the thermal resistance and thermal mass of the water and glass panes are negligible.

$$q_I = U(\theta_e - \theta_i) + U_w(\theta_{IN} - \theta_i) + A_I i_0,$$ (6)

where $U_w$ is the thermal transmittance between the interior and the water chamber, $U$ is the thermal transmittance of the glazing, and $\theta_{IN}$ is the inlet temperature of the fluid that enters the WFG. Taking into account Equations (3), (5), and (6), the expression for the total indoor heat flux, $q$, is:

$$q = U(\theta_e - \theta_i) + U_w(\theta_{IN} - \theta_i) + g i_0,$$ (7)

where:

$$U = \frac{U_i\,U_e}{\dot{m}c + U_e + U_i},$$ (8)

$$U_w = \frac{U_i\,\dot{m}c}{\dot{m}c + U_e + U_i}.$$ (9)

Water flow glazing has the potential to modify its thermal properties by changing the mass flow rate per unit of surface, $\dot{m}$. $U$ and $U_w$ depend on the mass flow rate $\dot{m}$, which is considered uniform within the glass pane. The product $\dot{m}c$ is the capacity of the water flow to absorb energy. $U_i$ and $U_e$ are thermal transmittances that can be obtained using the convective heat coefficients, $h_e$, $h_i$, $h_g$, and $h_w$. $U_e$ is the thermal transmittance from the water chamber to the outdoors. $U_i$ is the thermal transmittance from the water chamber to indoors. The definition of the $g$ factor for WFG is the same as stated in Equation (3) for double glazing. However, the expression for determining $A_I$ depends on the flow rate.

$$A_I = \left(\frac{U_i}{\dot{m}c + U_e + U_i}\right)A_v + A_i.$$ (10)

In a WFG with 2 glass panes and a water chamber:

$$\frac{1}{U_e} = \frac{1}{h_e} + \frac{1}{h_w},\tag{11}$$

$$\frac{1}{U_i} = \frac{1}{h_i} + \frac{1}{h_w}.\tag{12}$$

The absorptance, $A_v$, depends on the energy absorbed by the glass panes and by the water:

$$A_v = A_1\left(\frac{U_e}{h_e}\right) + A_2\left(\frac{U_i}{h_i}\right) + A_w,\tag{13}$$

where $A_1$ and $A_2$ are the absorptances of the outdoor and the indoor glass pane, respectively, and $A_w$ is the absorptance of the water chamber. The Equations (14) and (15) determine absorptance $A_i$ and $A_e$.

$$A_i = A_2\left(1 - \frac{U_i}{h_i}\right),\tag{14}$$

$$A_e = A_1\left(1 - \frac{U_e}{h_e}\right).\tag{15}$$

Figure 3 illustrates two panes of glass that make up a water chamber with an ascending flow of water, where $\theta_{IN}$ is the inlet temperature of the water chamber and $\theta_{OUT}$ is the temperature at the outlet of the water chamber. Out of the incoming solar radiation ($i_0$), the $g$ factor represents the percentage of the heat that ends up going through the glazing. $T$ is the transmittance of the glass. The absorbed energy is divided into three fractions. A constant part of the secondary internal heat transfer factor, $A_i$, is transferred indoors. Another constant part of the secondary external heat transfer factor, $A_e$, is transferred outdoors. Both, $A_i$ and $A_i$, do not depend on the flow rate. The remaining net absorptance of the water chamber, $A_v$, is transferred indoors, ($A_{v\ int}$), outdoors, ($A_{v\ ext}$), and transported by the water chamber ($A_{v\ water}$). The three factors depend on the flow rate. Using the hypothesis stated at the beginning of this section, the thermal profile inside each glass pane does not change with time. T denotes the transmittance of the glazing. The spectral properties depend on the incoming solar radiation, $i_0$.

The $g$ factor and transmittance of double glazing are constants. When it comes to WFG, these values can be actively controlled by modifying the flow rate, $\dot{m}$.

$$g = T + A_I.\tag{16}$$

Introducing the value of $A_I$ from Equation (10):

$$g = T + A_i + A_{v\ int}.\tag{17}$$

$A_{v\ int}$ impacts the g factor. Equation (8) shows that $A_{v\ int}$ is affected by the flow rate.

$$A_{v\ int} = \left(\frac{U_i}{\dot{m}c + U_e + U_i}\right)A_v.\tag{18}$$

The g factor for WFG is:

$$g = \left(\frac{U_i}{\dot{m}c + U_e + U_i}\right)A_v + A_i + T.\tag{19}$$

If the flow rate is much higher than the sum of indoor and outdoor thermal transmittances ($\dot{m}c \gg U_e + U_i$), the values of $g$ and $U$ are denoted with the superscript ON. The g factor of Equation (20) becomes:

$$g^{ON} = A_i + T.\tag{20}$$

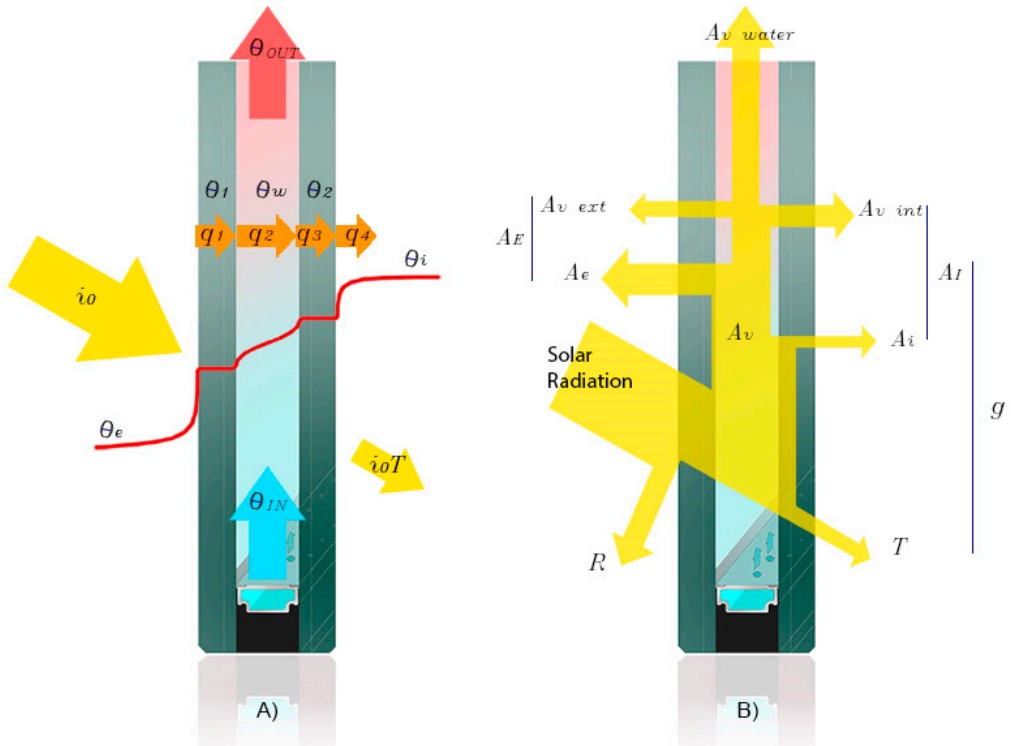

**Figure 3.** (**A**) Temperature distribution in a WFG and description of heat fluxes and temperatures of layers at a specific height of the envelope. (**B**) Distribution of solar radiation on the glazing. It is divided into the reflectance ($R$), the energy transmittance ($T$), the constant part of the secondary external and internal heat transfer factor ($A_e$ and $A_i$ respectively), and the net absorptance of the water ($A_v$).

The condition $\dot{m}c \gg U_e + U_i$ establishes the range of validity for the ON values. By using the thermal parameters detailed above, this condition becomes $\dot{m} > 0.0073$ kg/m$^2$s ($\dot{m} > 0.4$ l/m$^2$min) for double glazing with a water chamber. Figure 4 shows the $U$ value from Equation (8) and the $g$ factor from Equation (19) for different spectral parameters. $U$ ranges from 4.8 W/m$^2$ K when there is no flow ($U^{OFF}$) to 0.66 W/m$^2$ K when $\dot{m} = 0.039$ Kg/m$^2$s. The g-factor depends on $\dot{m}$, but also on the spectral properties of the glass.

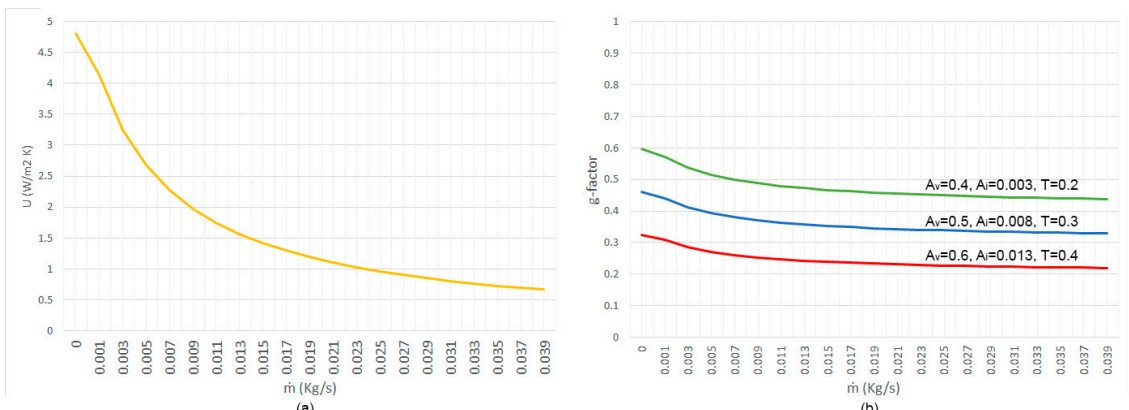

**Figure 4.** (**a**) U value of WFG depending on the mass flow rate, $\dot{m}$. (**b**) g-factor of WFG depending on the mass flow rate, $\dot{m}$, with three different glazing types.

The superscripts OFF and ON are used when the flow rate is stopped and when it is very high, respectively. When the flow rate is high ($\dot{m}c \gg U_e + U_i$), the Equation (7) becomes:

$$q = U_w^{ON}(\theta_{IN} - \theta_i) + g^{ON}i_0, \tag{21}$$

and Equation (9) becomes:

$$U_w^{ON} = U_i. \tag{22}$$

Considering the water chamber as an open system, the power per unit of surface that a flow of water can transport is:

$$P = \dot{m}c(\theta_{OUT} - \theta_{IN}). \tag{23}$$

The heat flux that goes through the two glass panes and the water is shown in Figure 3 and can be split into $q_1, q_2, q_3, q_4$. The unknown $\theta_{OUT}$ is obtained by solving the Equations (25)–(31).

$$q_2 = q_1 + A_1\, i_0\, , \tag{24}$$

$$q_3 = q_2 + A_w\, i_0\, + \dot{m}c(\theta_{IN} - \theta_{OUT}), \tag{25}$$

$$q_4 = q_3 + A_2\, i_0\, , \tag{26}$$

$$q_1 = h_e\, (\theta_e - \theta_1), \tag{27}$$

$$q_2 = h_w\, (\theta_1 - \theta_w), \tag{28}$$

$$q_3 = h_w\, (\theta_w - \theta_2), \tag{29}$$

$$q_4 = h_i\, (\theta_2 - \theta_i). \tag{30}$$

Thus, the outlet temperature of the water chamber is:

$$\theta_{OUT} = \left( \frac{A_v\, i_0\, + U_i\, \theta_i + U_e\, \theta_e + \dot{m}c\theta_{IN}}{\dot{m}c + U_e + U_i} \right) \tag{31}$$

where $\dot{m}$ is the mass flow rate per unit of area, $c$ is the specific heat of water, $A_v$ is the net absorptance of the water chamber, $i_0$ is the direct sun radiation, $\theta_{IN}$ is the temperature of the inlet water, $U_i$ is the interior thermal transmittance, and $U_e$ is the exterior thermal transmittance. These last two thermal transmittances $U_i$ and $U_e$ measure the heat transfer between the water chamber and indoors and outdoors, respectively. $(A_v i_0)$ is the sum of the energy absorbed by the water chamber and the energy transferred by convection because of the glass panes absorption. $A_v$ comes from Equation (13). It measures the fraction of energy that is absorbed by the water. Regardless of the flow rate, the maximum amount of energy that the water chamber can absorb is $A_v i_0$. When $\dot{m}$ reaches a specific value, $g_w$ becomes negligible and the absorbed energy is not transferred indoors. Therefore, the absorbed energy, $A_v i_0$, is transported by the water chamber.

### 2.3. Glass Selection

The simulation tool, used to select the thermal and spectral properties, is based on previous papers that have shown a mathematical model to predict the performance of WFG, along with other radiant systems [17,42]. The selection of the glazing impacts the design of the HVAC system of buildings. This subsection aims to select the WFG parameters by using the simulation tool. The four tabs include:

1. Energy balance considerations based on the location, including potential sun energy.
2. Spectral properties of glass panes and coatings.
3. Thermal simulator of the WFG modules.
4. Thermal simulator of simplified rooms combining WFG and non-transparent walls, roofs, and floors.

Designing a WFG facade starts with the analysis of the potential solar energy. A few factors include location, usage, and the orientation of different facades. This information is shown in graphs that represent the energy balance considerations for a particular day, week, month, or year. To accomplish the energy balance between the demand and the energy absorption, the dynamic $g$ factor and thermal

transmittance are considered. The wavelength of solar radiation, the orientation, and the angle are the factors that affect spectral properties of glass, such as transmittance, reflectance, and absorptance. The second phase is to define the simulation inputs, such as exterior temperature, direct and diffuse solar radiation, the selected flow rate, and inlet temperature in the WFG. The outputs of the simulation tool are the indoor temperature, the outlet temperature in the WFG, and the water heat gains calculated over some time. Finally, the simulation tool allows the user to introduce the dimensions of rooms, the thermal properties of opaque facades, and the interior absorption properties of walls, roofs, and floors. The spectral properties of the glass have been taken from the software Optics and Window [43], which are based on the International Glazing Database (IGDB). Manufacturers provide optical and spectral data of their products with standards of quality. The composition of the WFG comprises a laminated glass pane of 6 + 6 mm, a water chamber of 16 mm, and another laminated glass pane of 8 + 8 mm. The glass panes selected were Planiclear 6 and 8 mm thick (manufactured by Saint Gobain), and the Poly-Vinyl Butyral (PVB) layers were standard (1 × 0.38 mm). Table 1 presents the thermal and spectral parameters of the WFG, along with the Reference glazing.

**Table 1.** Thermal and spectral properties of glazing.

| | $A_1$ | $A_2$ | $A_w$ | $U$ | $U_w^{ON}$ [1] | $g$ | $g^{ON}$ [1] |
|---|---|---|---|---|---|---|---|
| **Reference Glazing** | 0.585 | 0.037 | | 2.6 | - | 0.67 | - |
| **WFG** | 0.585 | 0.037 | 0.429–0.524 | 0.762 | 5.802 | - | 0.544 |

[1] These values have been obtained with the hypothesis: $\dot{m}c \gg U_e + U_i$.

## 3. Results

This section aims to validate the model presented in Section 2.2 with real data from the prototype described in 2.1. Regarding the thermal parameters, the coefficients $h_i$ and $h_e$ are taken from the European Standard [39,40], being $h_i$ = 10 W/m$^2$ K and $h_e$ = 25 W/m$^2$ K for horizontal glazing with heat flux going upwards. The value of $h_w$ is the result of using Equation (11) of Reference [17], taking into account the Nusselt number of 7.541. The thermal resistance of a 16 mm water layer is 0.026 m$^2$K/W. Finally, $h_w$ = 452 W/m$^2$ K.

The values of $U_e$ and $U_i$ are obtained with Equations (11) and (12).

$$(1/U_e) = (1/h_e) + (1/h_w) = 1/10 + 1/452 = 0.102; U_e = 9.80.$$

$$(1/U_i) = (1/h_i) + (1/h_w) = 1/25 + 1/452 = 0.042; U_i = 23.69.$$

Typical values for $A_v$ are given in previous articles [36] depending on the sum of the absorptances of each layer, which makes up the glass pane. A range of values from 0.331 to 0.429 has been considered in this case study. The flow rate should be high enough to transport the absorbed energy. Then, if $\dot{m}c$ is much higher than $U_i + U_e$, the proportion of the absorbed energy that is transferred indoors, $A_I$, and outdoors, $A_E$, is minimized.

In this section, real data of monitoring temperatures and power efficiency of the model described in Section 2.2 are presented. By analyzing the performance of the WFG cabin, the energy strategy may be refined or improved, achieving energy savings. Hence, the power performance of the WFG envelope is obtained by measuring the inlet and outlet temperature and the flow rate of each facade.

### 3.1. Data from the Prototype

Real data, obtained from the prototype in three consecutive days of Summer, allows the comparison between the WFG cabin and the Reference cabin. It is essential to highlight that the prototype is not designed to assure comfort temperature inside but to evaluate the potential solar energy that WFG envelopes can absorb for Energy Management strategies. When the sun radiation impinges in the glazing units, an amount of energy is absorbed in the water chamber. This harvested energy can be

distributed through the rest of the glazing facades of the cabin by homogenizing temperature gradients. Besides, energy surplus can be stored in the geothermal boreholes heat exchangers.

Figure 5 shows the curves of the outdoor temperature (printed in grey), the indoor temperature of the WFG cabin (printed in cyan), and the indoor temperature of the Reference cabin (printed in red). As can be seen, the peak outdoor temperature reaches 31.5 °C, and the peak indoor temperature in the WFG cabin rises to 38.5 °C. In contrast, the peak indoor temperature of the Reference cabin amounts to 53.5 °C. Hence, the temperature inside the WFG cabin is reduced in 15 °C in relation to the Reference cabin as a direct consequence of the fact that the solar energy absorbed by the water chamber is transferred to the ground before being transmitted indoors. Therefore, although the composition of the glass facade is the same for both prototypes, the solar factor of the WFG cabin when the system is operating at the design flow rate $\dot{m}$ = 0.9 l/min m$^2$, is reduced to 0.54, compared to the 0.67 provided by the manufacturer, as it is shown in Table 1.

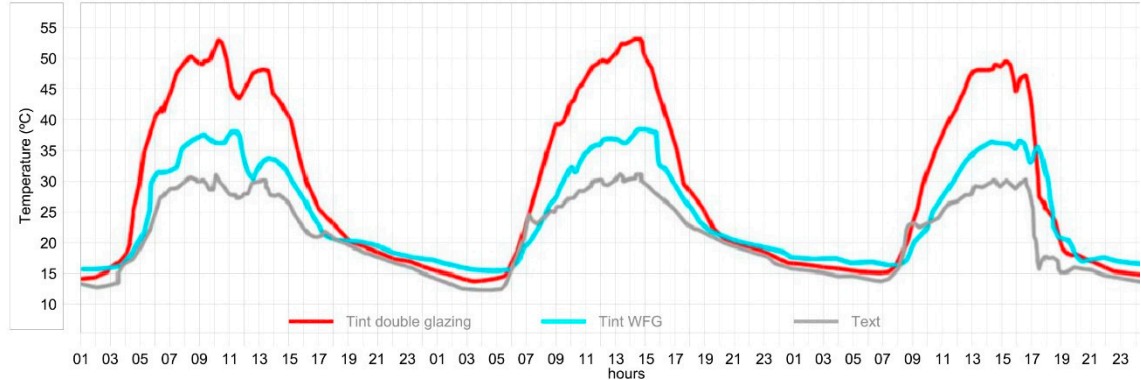

**Figure 5.** Measured data of the outdoor temperature and the indoor temperature of the WFG cabin and Reference cabin. Sample summer days: 23/08/2010, 24/08/2010, and 25/08/2010.

Furthermore, the effect of thermal inertia derived from geothermal boreholes contributes to cushioning the temperature curve of the WFG prototype compared with the Reference one, as it is shown in the graphs of Figure 4. Even, the benefits of the effect of thermal inertia are also reflected when the peak temperature of the WFG prototype is one hour behind from the Reference cabin.

Figure 6 shows the inlet and outlet temperatures of the fluid taken from the horizontal glazing. As can be seen in the graph, when the inlet temperate is lower than the outlet temperature, the solar energy is absorbed in the water chamber over the daylight hours and transferred to the ground through the borehole heat exchangers. Likewise, over the night, the temperature of the glazing is cooled down, and the inlet temperature is higher than the outlet temperature.

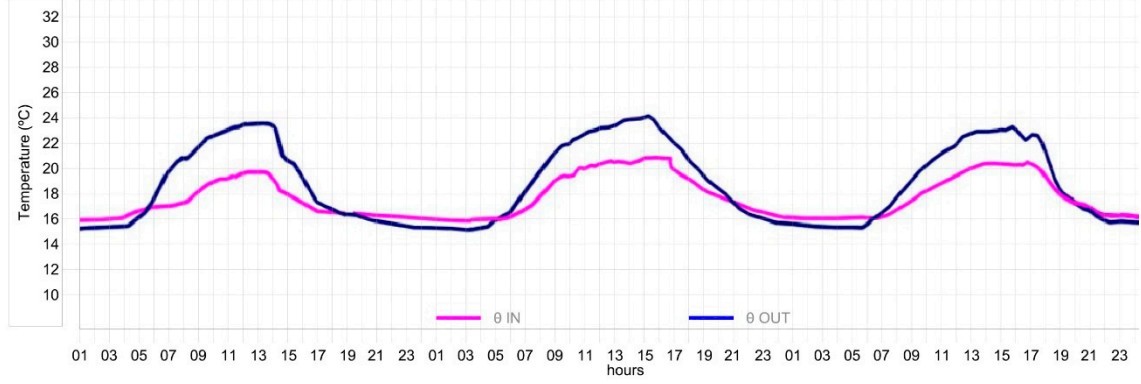

**Figure 6.** Measured data of the inlet and outlet temperature of the WFG cabin. Sample summer days: 23/08/2010, 24/08/2010, and 25/08/2010.

### 3.2. Validation of the Mathematical Model

The design mass flow rate of the water flow glazing was taken from real data from the prototype. The design mass flow rate $\dot{m}$ was 3.2 L/min through 3.5 m$^2$ of horizontal WFG, so the steady value $\dot{m}$ = 0.9 L/min m$^2$ (0.015 kg/s m$^2$). This value has been introduced in the tool, and it is high enough to validate the hypothesis in Equation (21). The heat capacity of the fluid, a mixture of water and glycol, is $c$ = 3600 J/kg K. The boundary conditions were the same in both the mathematical model and the real prototype. The inlet temperature of the fluid was taken from real data, and the mass flow rate per unit of surface used in the simulation was $\dot{m}$ = 0.015 kg/m$^2$s.

A pyranometer installed on the roof of the prototype measures global solar irradiance. Data acquisition is made through the monitoring system. Figure 7 shows the irradiance curve corresponding to the selected days (23/08/2010, 24/08/2010, and 25/08/2010). As it is shown, the selected days were clear. When the day is clear, direct beam radiation predominates over diffuse radiation. However, when the day is cloudy, almost all the irradiance is diffuse. Likewise, the simulation curve (printed in cyan) shows the typical irradiance curve reaching maximum levels of around 800 W/m$^2$. Over the first hours in the morning (from 09:00 to 13:00 h), and in the late afternoon (after 18:00 h), the roof is shaded and the irradiance is mainly diffuse reaching values around 200 W/m$^2$. However, throughout the central hours of the day, the roof is exposed to direct solar radiation.

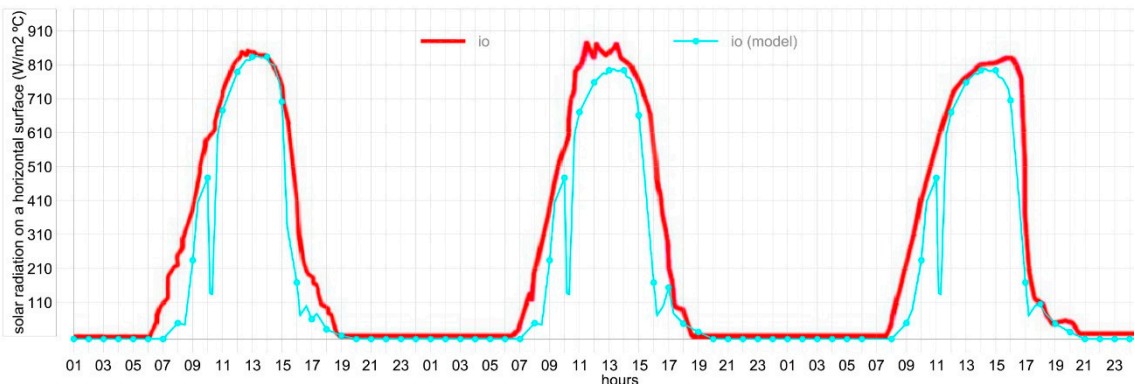

**Figure 7.** Solar irradiance. Comparison results of simulation data and measured data of the solar irradiance of the prototype. Sample summer days: 23/08/2010, 24/08/2010, and 25/08/2010.

Figures 8–10 show the validation of the prototype and the Software Tool using real data. These figures show the main results of three days in August (23/08/2010, 24/08/2010, and 25/08/2010) where the measured curves (printed in a continuous line) replicate the simulation curves (printed in a continuous line with marked dots) in all the cases.

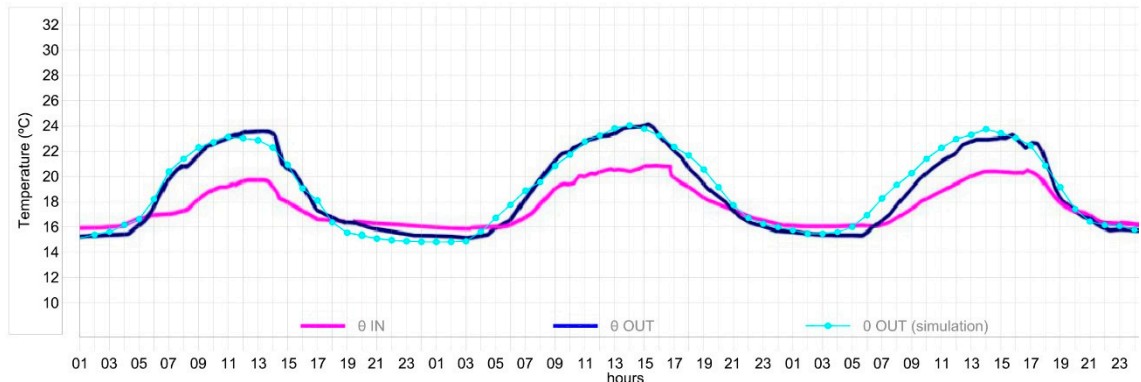

**Figure 8.** Comparison between measured data of the inlet and outlet temperature of the WFG cabin and results from simulation data. Sample summer days: 23/08/2010, 24/08/2010, and 25/08/2010.

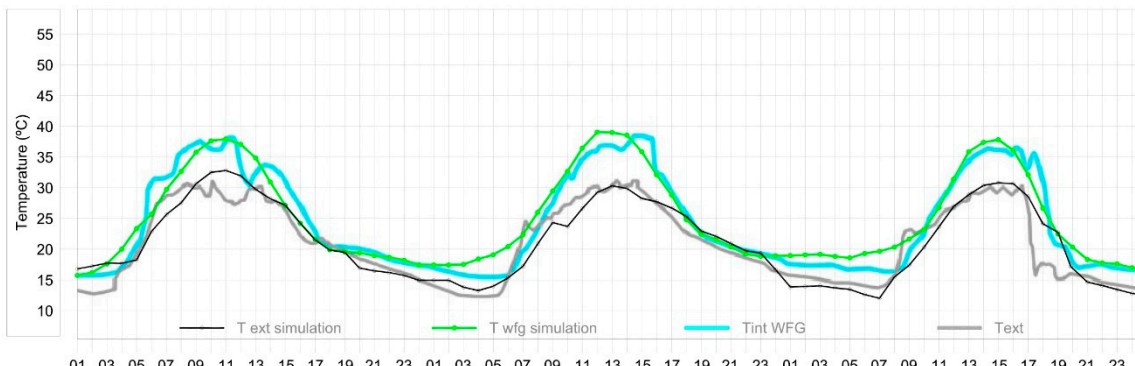

**Figure 9.** Performance of WFG. Room indoor temperature in the WFG cabin and outdoor temperature. Comparison between simulation results calculated with the tool and real data from the prototype. Sample summer days: 23/08/2010, 24/08/2010, and 25/08/2010.

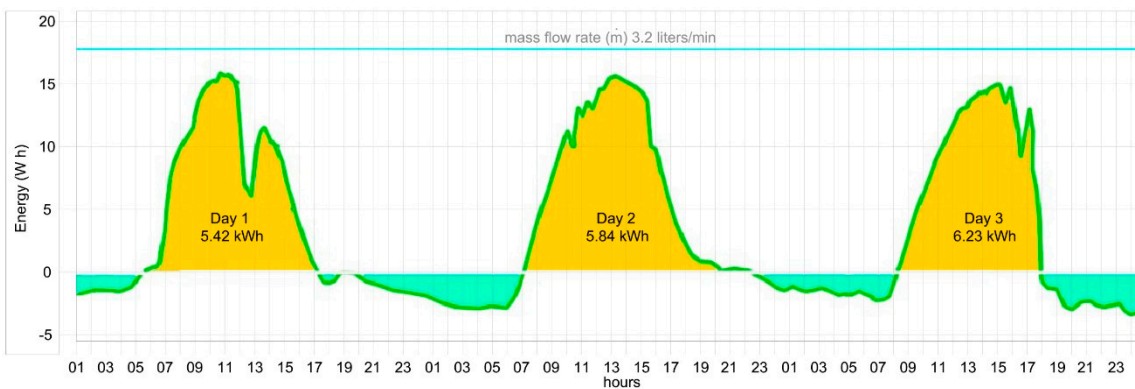

**Figure 10.** Energy absorbed in the horizontal glazing. Area of glazing = 3.5 m$^2$; mass flow rate = 3.2 L per minute. Sample summer days: 23–26 August.

## 4. Discussion

Double glazing has constant $g$ and $U$ values. In WFG, these values can be actively controlled or modified by the flow rate $\dot{m}$. A WFG allows controlling the incoming solar energy into a building by changing its flow rate. A primary system should be designed for the building to manage this absorbed energy. The studied dynamic simulation tool assessed the energy performance of new building technologies that are not implemented yet in commercial building energy simulation software. Real data obtained in the summer period over the selected days (23/08/2010, 24/08/2010, and 25/08/2010). These days allow validating the simulation tool, considering the highest potential of the WFG over the summer, when the sun irradiance reaches maximum levels. The outdoor temperature and solar irradiance vary during the day and thermal performances are functions of time. Hence, transient behavior is expected.

Figure 7 shows how the measured curves of the solar irradiance (printed in a continuous line) replicate the simulation curve (printed in a continuous line with marked dots), following the same shape and remarkably similar values slightly over 800 W/m$^2$, which would be expected for the summer period in this location. This prototype is simulated in the actual location and the weather file is the standard EPW file (EnergyPlus Weather).

### 4.1. Comparison between the WFG Cabin Data and the Simulation

Concerning the temperatures of the WFG cabin, Figure 9 shows that real data curves do not precisely replicate the behavior according to the simulation both in terms of the interior and exterior temperatures. The interior temperature of the WFG cabin is undoubtedly the most critical variable to validate with the built prototype since it can be affected by infiltrations or thermal bridges coming

from the aluminum profiles or the connection between the slab and the glazed walls. These would explain the temperature oscillations in the curves, especially on 23 August. However, in both measured and simulated curves, the interior peak temperature remains below 40 °C. In contrast, at minimum temperatures, it is the actual data curve that registers a peak value of less than 14 °C, compared to the 15.5 °C provided by the simulation tool. Likewise, the exterior temperature presents very slight differences between the real data curve and the simulated curve. Some oscillations in the former curve between 09:00 and 13:00 on 23 August and between 17:00 and 19:00 on 25 August would indicate that although the days were bright and sunny, some shadow coming from the prototype itself could be slightly influencing the exterior temperature probe. The difference between the measured value and simulation results is the Mean Error (ME). The Mean Percentage Error (MPE) is calculated to validate the simulation results with the real data. The total number of measurements was n = 10,167. Both values are shown in Equations (32) and (33).

$$ME = \frac{1}{n} \sum_{i=1}^{n} |T_{Si} - T_{Ri}|, \tag{32}$$

$$MPE = \frac{1}{n} \sum_{i=1}^{n} \left| \frac{T_{Si} - T_{Ri}}{T_{Ri}} \right| 100. \tag{33}$$

where $T_{Si}$ is the simulated value, and $T_{Ri}$ is the measured value. By computing ME and MPE, the sample summer days: 23/08/2010, 24/08/2010, and 25/08/2010, MEs and MPEs of the outlet temperature, $\theta_{OUT}$, were lower than 0.3 C and 2.1%, respectively. When it comes to the indoor temperature, $\theta_i$, the ME was 0.7, and the MPE was 2.8%. The reason for this might be the uncertainties about infiltration, which might affect the indoor temperature. Predictions of $\theta_{OUT}$ were more accurate because the boundary conditions were more suitable to predict.

### 4.2. Energy Management of the WFG Cabin

When analyzing the WFG cabin, the glazing is part of a room, and the thermal problem of the glazing is coupled with the thermal problem of the room, consequently, the indoor temperature is unknown, and it should be obtained at the same time as the glazing temperature profile. Furthermore, when the system is circulating, the flow rate is set to the design flow rate 0.9 l/min m$^2$ and the inlet temperature, $\theta_{IN}$, varies between 16 °C and 20 °C. The outlet temperature, $\theta_{OUT}$, is also an unknown and should be obtained, as it is shown in Equation (31). Likewise, when the system is stopped, the flow rate is set to zero, and the outlet temperature of the water chamber, $\theta_{OUT}$, is called the stagnation temperature. These outputs are the water heat gain or thermal power of the WFG. Hence, as it is shown in Figure 8, the outlet temperature ranges between 24.5 °C during the day and 15 °C during the night, matching with the simulation curve. Both measured and simulated curves reveal that when the inlet temperature is above the outlet temperature, the WFG is cooled down. This effect takes place during the night and allows to fully grasp the energy strategy that the WFG envelope follows, absorbing solar energy in the water chamber and transferring that heat into the ground through the borehole heat exchangers.

Using Equation (23) and the values taken from Figure 8, the daily energy per unit of surface that the flow of water can transport was calculated. WFG stops this energy from heating up the prototype. It explains the difference in temperatures between the WFG cabin and the Reference cabin. This energy is released through the borehole heat exchangers. Figure 10 shows the energy absorbed by the horizontal WFG. The daily average absorbed energy was 5.8 kWh in 3.5 m$^2$, so the ratio of energy per area is 1.66 kWh/m$^2$. Table 2 illustrates the energy balance between energy absorption and the electric consumption of the circulation water pumps. The circulation pump was working 24 h per day, and its steady power was 25 W. The average daily use of the circulation pump to provide the WFG

with a flow rate of 0.9 L per minute was 0.58 kWh, which represents a 10% of the energy absorbed by the water. The net energy savings were 1.49 kWh/m$^2$.

**Table 2.** Energy balance of the horizontal WFG.

|  | Absorbed Energy, kWh (Day Time) | Dissipated Energy, kWh (Night Time) | Water Pump, kWh (24 h) |
|---|---|---|---|
| Day 1 | 5.42 | 1.47 | 0.58 |
| Day 2 | 5.84 | 1.55 | 0.58 |
| Day 3 | 6.23 | 1.51 | 0.58 |

Table 3 compares the final energy consumption of a heat pump with a seasonal coefficient of performance (SCOP) of 3 with the energy consumed by the water pump. The primary energy factor (PEF) from final energy (FE) to non-renewable final energy (NRFE) was 1.954. The factor of CO2 emissions for electricity was 0.331 [44].

**Table 3.** Final energy analysis. Sample summer days: 23/08/2010, 24/08/2010, and 25/08/2010.

|  | WFG Absorbed Energy (3-Day Time) | Water Pump (3-Day Time) |
|---|---|---|
| Energy consumption | 17.49 | 1.74 |
| SCOP | 3 |  |
| FE consumption (Kwh) | 5.83 | 1.74 |
| NRFE consumption (Kwh) | $5.83 \times 1.954 = 11.39$ | $1.74 \times 1.954 = 3.39$ |
| CO2 emissions (KgCO2) | $5.83 \times 0.331 = 1.93$ | $1.74 \times 0.331 = 0.57$ |

WFG allows considering the envelope as a part of the HVAC system. The building envelope is the first shield that receives solar radiation and protects the building against the cold and hot weather. Traditional curtain walls use passive measures, such as coatings and gas cavities. WFG can be used as a component of an active energy management strategy. This article has not focused on analyzing the behavior of the borehole heat exchangers. Further development of this research may analyze the design of an appropriate shallow geothermal system, taking into account the predicted energy absorbed by WFG envelopes.

*4.3. Cost Considerations*

Borehole heat exchangers are considered low valued and environmentally sustainable energy sources for heating and cooling of buildings that contribute to reducing the primary energy demand. In addition, borehole heat exchangers are considered low exergy (LowEx) systems that allow designing high-performance buildings using the freely available dispersed energy in the environment [45].

The costs of the system made up of WFG and borehole heat exchangers are high compared to passive glazing systems. The cost of boreholes ranges between € 30 and € 50 per meter of drilling. The expected energy yield is 50 W per meter, depending on the characteristics of the soil [11,12]. Figure 10 shows that the average daily energy absorbed by the water is 5.8 kWh over 10 h. The borehole length to dissipate 5.8 kWh is 11.6 m, so the extra cost of this system is € 580 for a 3.5 m$^2$ of WFG. Table 3 shows that the difference in NRFE compared with an air-to-air heat pump is 2.66 Kwh/day. The average price of electricity in Spain is 0.2165 €/Kwh [46], and the daily savings are 0.58 €/day, so the expected return of investment time is less than 3 years.

Furthermore, the commissioning process of the studied system includes preventive and predictive maintenance protocols. The commissioning process is an effective method to ensure that buildings reach their operating potential and that advanced components and systems arrive at their technical specifications. The relationship between investment in commissioning and the cost reduction of the entire project is a topic for further research.

## 5. Conclusions

Dynamic envelopes, such as WFG, supplied by renewable energy sources like boreholes heat exchangers, can adapt to the building environment instead of creating an insulated barrier. This article has shown that it is possible to simulate dynamic parameters for WFG, such as $g$ factor and $U$ value. Besides, it appears a second thermal transmittance, $U_w$, associated with the flow of water. Section 2.2.1 provided a mathematical model to calculate $U$ and $g$-factor as dynamic properties of double WFG. $U$ ranges from 4.8 W/m$^2$ K when the mass flow rate $\dot{m} = 0$ ($U^{OFF}$) to 0.66 W/m$^2$ K when $\dot{m} = 0.039$ Kg/m$^2$s. The $g$-factor depends on $\dot{m}$ but also the spectral properties of the glass. The $g$-factor can be modified up to 60% by changing the mass flow rate from $\dot{m} = 0$ ($g^{OFF}$) to $\dot{m} = 0.039$ Kg/m$^2$s.

The potential of WFG as a means to reduce the cooling loads in summer has been tested. A prototype with WFG in facades and the roof, connected to borehole heat exchangers, was built along with another prototype with the same dimensions and made of traditional double glazing. The effect of thermal inertia derived from geothermal boreholes contributes to flatten the temperature curve of the WFG prototype compared with the Reference one. Besides, the non-renewable final energy (NRFE) consumption that is needed to compensate for the absorbed solar radiation is 11.39 Kwh in 3.5 m$^2$ of horizontal WFG. In contrast, the NRFE required to run the circulating pump over the same period is 3.39 Kwh. It represents 70% of NRFE savings. The difference in CO2 emissions between the two options is 1.36 KgCO2. It represents a 70% reduction in CO2 emissions.

The reliability of the mathematical model to simulate the behavior of the prototypes was tested by developing real prototypes. In this paper, the results of the experimental data were compared with the simulation. Two parameters were simulated in the WFG cabin: the indoor air temperature and the outlet temperature in the WFG roof panels. The mean error (ME) and the mean percentage error (MPE) were measured.

- The ME of the outlet temperature, $\theta_{OUT}$, was 0.29 C, and the MPE was 2.1%.
- The ME of the indoor temperature, $\theta_i$, was 0.7, and the MPE was 2.8%.

Finally, a number of new topics can be explored in future studies. The results of this article may lead to the following future research suggestions:

- The performance of WFG in the wintertime.
- The effect of WFG in comfort by analyzing the mean radiant temperature (MRT).
- The study of different energy generation systems in buildings (fuel-based or heat pumps) and its potential for energy savings using WFG envelopes.

**Author Contributions:** Conceptualization, B.M.S., F.d.A.G., and J.A.H.R.; methodology, B.M.S. and F.d.A.G.; software, J.A.H.R.; formal analysis, R.-A.G.-L and F.d.A.G.; data curation, F.d.A.G. and J.A.H.R.; writing—original draft preparation, B.M.S., F.d.A.G., and J.A.H.R.; writing—review and editing, D.P. and B.M.S; visualization, D.P., B.L.A., and R.-A.G.-L.; supervision, J.A.H.R. and B.L.A.; project administration, F.d.A.G. and R.-A.G.-L.; funding acquisition, F.d.A.G. and R.-A.G.-L. All authors have read and agreed to the published version of the manuscript.

**Funding:** This article has been funded by the KSC Faculty Development Grant (Keene State College, New Hamshire, USA). The authors wish to thank CEU San Pablo University Foundation for the funds dedicated to the Project CEU-Banco Santander (Ref: MVP19V14) provided by CEU San Pablo University and financed by Banco Santander.

**Acknowledgments:** This work was supported by program Horizon 2020-EU.3.3.1: Reducing energy consumption and carbon footprint by smart and sustainable use, project Ref. 680441 InDeWaG: Industrialized Development of Water Flow Glazing Systems. The authors wish to thank the municipality of Peralveche, Spain, for its generous support.

**Conflicts of Interest:** The authors declare that they have no conflict of interest.

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
