# Peer review of "Application and Validation of a Dynamic Energy Simulation Tool: A Case Study with Water Flow Glazing Envelope"

_energies, doi:10.3390/en13123203_

Round 1

Reviewer 1 Report

The work presented has a high quality in the proposed study, objectives, methodology and results presentation.

The paper is addressing a contemporary research based on innovative solutions for buildings, namely the use of water flow glazing and borehole heat exchangers. The study presents numerical model validated with experimental results.

Highly recommendation for publication.

Author Response

We want to thank the reviewers and the Editor for their detailed comments and suggestions for our manuscript. We firmly believe that their comments have been beneficial to identify essential areas that required improvement. After completion of the edition, the revised manuscript has benefited from such an improvement in its overall presentation and clarity.

Some changes have been made in the manuscript to fix typos and clarify aspects of the Research.

Reviewer 2 Report

REVIEWER ANSWER

This paper aims to make a dynamic model of a water flow glazing solution for buildings envelope in order to incorporate it in libraries for further analyses. It is based on the data taken from a prototype with WFG connected to borehole heat exchanger. The work only focuses in the envelope performance and borehole HX is leaved for later study.

The article is written in a clear form and is easy to follow. The content adapts to the aim of the energies journal.

I found the core of the article very interesting and promising but, nevertheless, there are some aspects I would like to remark and some reviews I would like to suggest:

  • To begin with, I would modify the tittle a little bit by erasing the “borehole heat exchanger” specification. After all, no analysis is carried on with the borehole HX part and can be a bit confusing: APPLICATION AND VALIDATION OF A DYNAMIC ENERGY SIMULATION TOOL: A CASE STUDY WITH WATER FLOW GLAZING ENVELOPE.
  • Because of the same reason, I found the information given about borehole heat)exchangers additional (lines 44-66). The information is right, but, the state of art should be more focus on WFG and dynamic modelling. Notwithstanding this point, I would not delete but shorten that paragraph.
  • In line 50, “This solution is optimum in cold countries when the systems allows heat injection into the ground in summer” is written. Is that the case of Madrid?
  • In line 56 GSHP are included. Is your system working with geothermal heat pumps? (if the answer is positive, I did not found it in any place).
  • I suggest to incorporate a subtitle, in line 68, since you pass from GSHP to building envelope. The introduction guiding thread would be easier to follow.
  • Just a comment: sometimes "facade" is written and other times "façade". Please, homogenize the writing. (i.e. line 82 “facade”)
  • In line 87: “Releasing the excess of heat in the ground during the summer allows to increase the thermal inertia of a light and transparent envelop”. What happens on wintertime? Is the system turned off? (same question appears in in line, 141 “On the other hand, the ground temperature regenerates during the summer […]”)
  • In lines 92-96 I will include that “active water-flow glazing” requires extra energy consumption compared to “traditional construction elements” (note: I realize that in line 118 “The only additional energy source was the electricity needed to run the circulation water pumps.”, so I suggest to bring it forward to line 96)
  • Once again I suggest to include a subtitle in line 105: “Objectives” or “Innovations”. Something that facilitates the reading.
  • In line 113: “The steady-state model is not accurate when it comes to dynamic forms of heat transfer, additionally affected by indoor HVAC systems” is written. I totally agree. Nevertheless, is not your model based on a steady state model? 

The same question appears to me in line 240: “Using the hypothesis stated at the beginning of this section, the thermal profile inside each glass pane does not change with time”. Therefore, are you working with quasi-static condition (without considering inertias and previous situations) or in dynamic one?

  • In line 125 “[…] than the systems currently used” I will add: “which are, most of them, feed by fossil fuels”
  • In line 127: “The empirical tests were carried out for several months under variable weather conditions and in a free-floating indoor temperature regime.” what does "several months" means? how many months did you use?

Besides, I realize that the validation is done with only three days, as written in line 338: “Real data, obtained from the prototype in three consecutive days of Summer, allows the comparison between the WFG cabin and the Reference cabin” 

And the same question arises in line 169 “Weather data was recorded for one year. The temperature inside the prototypes was measured at the central line below the roofs”. Which is the analysis period?

  • Just a suggestion, Figure 1 is explained during the lines 158-161. It will be easier to follow the text and the figure if, during the text, the same numbering is written: i.e. borehole HX (4)...
  • just a comment: when units are written, some times the "2" is a sub-index and other time as a number, (i.e. line 173: 3000 W/m2). Please, agree one typography
  • in line 181: “This study used the equations and the mathematical models from those articles, along with data from commercial software, to assess the performance of the prototypes described in section”. I do not understand which model is used. Are not you creating your own model of WFG?
  • I like Figure 3. Nevertheless, I will add to the A) explanation “on a specific height of the envelope”. This is to say: “Temperature distribution in a WFG and description of heat fluxes and temperatures of layers, on a specific height of the envelope”
  • Just a curiosity. For ?out achievement (eq. 31), why is not the “common internal forced convection” formula used? (?out = ?surf - ( ?surf - ?in)·exp(-h Asurf /mc)).

(This is maybe justified in previous bibliography as written in line 277: The simulation tool, used to select the thermal and spectral properties, is based on previous papers that have shown a mathematical model to predict the performance of WFG, along with other radiant systems).

  • Lines 270-275 are very interesting
  • From line 330-336 “objectives” of the article appears. Maybe can be included in the previous “objective” section
  • Lines 342-345 are very interesting
  • Line 354 is interesting as well as lines 365-366
  • During the lines 380-383, when comparing the real data with model outputs, which are the mean error and the maximum error?
  • I suggest to follow the same style of the previous Figures in the title of Figure 8. That is, I will add “Sample summer days: 23/08/2010, 24/08/2010, and 25/08/2010”.
  • In line 411 “Hence, unsteady behavior is expected.” Can it be that unsteady behaviour somehow avoided?
  • In line 415: “This prototype is simulated in the actual location and the weather file is the standard EPW file (EnergyPlus Weather).” why was not the real weather data used?
  • In table 2, and during the lines 451-455, heat is compared to electricity. Primary energy should be used instead absolute energy savings. What is more, if one wants to go deeply, exergy analysis would show that there are not benefits when electricity is used to heat or cool.

Besides, the energy savings should be extent to economic savings. After that, to go a little further, net economic savings should be calculated by considering the investment and maintenance costs compared to passive glazing systems.

  • Therefore, I miss the economic part: a subsection that accounts for the extra cost of incorporating boreholes and mechanical devices should be, at least, mentioned.

Reviewer 3 Report

Please add recent published water flow glazing articles in the introduction section

https://doi.org/10.1016/j.scs.2020.102152

https://doi.org/10.1016/j.renene.2018.04.038

line 139: Active- active

Line 146- spell error

Line 147: U-value= 6 or 0.6?

Replace the Figure 2 with a nicer one probably taken in a sunny day.

Line 189: unit is worng

Figure 4: what s text?

Figure 4-8 mostly shows the tempereature ad solar radiation data. How they are related to U-value and g-value calculation is not fully reflected here. Can these values are fully dynamic with the  single mass flow rate of variable mass flow rate?

Dynamic U-value and g value plot is possible ? If yes then add them up to show.

Section 4: discussion section is ok but need one conclusion section. Also it is interesting to know that changing mass flow rate U-value and g-value is changing.  However, switchable glazing performs this job by changing their transmission (SPD , PDLC and EC).  It better to add bit information about switchable glazing in the introduction and related them in the discussion also by explaining how this system is better than those system.

Round 2

Reviewer 2 Report

REVIEWER ANSWER

This paper aims to make a dynamic model of a water flow glazing solution for buildings envelope in order to incorporate it in libraries for further analyses. It is based on the data taken from a prototype with WFG connected to borehole heat exchanger. The work only focuses in the envelope performance and borehole HX is leaved for later study.

The article is written in a clear form and is easy to follow. The content adapts to the aim of the energies journal.

Considering the previous comments, the authors have clearly and satisfactorily answer point by point the suggestions. I now feel that the paper is ready to be published.

Author Response

We would like to thank the reviewers and the Editor for their detailed comments and suggestions for our manuscript. The revised manuscript has benefited from such an improvement in its overall presentation and clarity. Please find below a point-by-point description of how each comment is addressed within the manuscript.

Reviewer 3 Report

Most of the comments have been improved. In line 122-124 where you have mentioned about the switchable glazing add more other type for e.g. SPD one also. 

Also mention  limitation of water flow glazing and your work.

Author Response

We would like to thank the reviewers and the Editor for their detailed comments and suggestions for our manuscript. The revised manuscript has benefited from such an improvement in its overall presentation and clarity. Please find below a point-by-point description of how each comment is addressed within the manuscript. The original reviewer’s comments are in regular fontand our responses are in bold.

Most of the comments have been improved. In line 122-124 where you have mentioned about the switchable glazing add more other type for e.g. SPD one also. 

The following text and two more references have been added (lines 122-130):

Polymer Dispersed Liquid Crystal (PDLC) devices can adapt to different weather conditions, but their expected range of U value and g-factor is not as broad as expected for WFG [30]. SPD switches from transparent to colored due to the suspension of freely arranged particles between two glass panes. The electrochromic (EC) glass varies transmission and reflection parameters by electrical stimuli to mobile ions in the electrochromic layer. The solar heat gain coefficient of SPD and EC ranges from 0.5 in the transparent state to 0.06 in the non-transparent state [31]. The critical aspects remain in the high cost of the products and a lack of standardization in the manufacturing process [32].

[31] Allen, K., Connelly, K., Rutherford, P., Wu, Y. Smart windows—Dynamic control of building energy performance, Energy and Buildings, 2017, 139, 535-546 https://doi.org/10.1016/j.enbuild.2016.12.093.

[32] Casini, M., Smart windows for energy efficiency of buildings. International Journal Of Civil and Structural Engineering - IJCSE, 2015, 2 (1), 230-238 doi:10.15224/ 978-1-63248-030-9-56

Also mention  limitation of water flow glazing and your work.

The following text has been added (lines 153-156):

The challenges of WFG are related to the ability to produce large panels and their behavior in the long term. Although considerable efforts have been made in the industrialization process, there is still low market penetration because of the limited information between professionals and consumers.

Round 3

Reviewer 3 Report

https:/

Add the below for SPD U-value work
1.Measured overall heat transfer coefficient of a suspended particle device switchable glazing https://doi.org/10.1016/
2. Behaviour of a SPD switchable glazing in an outdoor test cell with heat removal under varying weather conditions

Author Response

We would like to thank the reviewers and the Editor for their detailed comments and suggestions for our manuscript. The revised manuscript has benefited from such an improvement in its overall presentation and clarity. Please find below a point-by-point description of how each comment is addressed within the manuscript. The original reviewer’s comments are in regular font and our responses are in bold.

Add the below for SPD U-value work

1.Measured overall heat transfer coefficient of a suspended particle device switchable glazing

https://doi.org/10.1016/

  1. Behaviour of a SPD switchable glazing in an outdoor test cell with heat removal under varying weather conditions

The following text has been added (lines 117-119), along with references 32 and 33.

The addition of double glazing to an SPD can vary the U value from 2.98 W/m2 K to 1.99 W/m2 K [32]. The heat transfer coefficient of SPD ranges from 5.02 W/m2 K and 5.2 W/m2 K for opaque and transparent states [33].

[32] Ghosh, A., Norton, B., Duffy, A. Measured overall heat transfer coefficient of a suspended particle device switchable glazing, Applied Energy, 2015, 159, 362-369. https://doi.org/10.1016/j.apenergy.2015.09.019.

[33] Ghosh, A., Norton, B., Duffy, A. Behaviour of a SPD switchable glazing in an outdoor test cell with heat removal under varying weather conditions, Applied Energy, 2016, 180, 695-706. https://doi.org/10.1016/j.apenergy.2016.08.029.